# Using patient-reported outcome measures during the management of patients with end-stage kidney disease requiring treatment with haemodialysis (PROM-HD): a qualitative study

Nicola Elzabeth Anderson [1,2,3] Christel McMullan [3,4]
Melanie Calvert [3,4,5,6,7] Mary Dutton,[1,2,3] Paul Cockwell,[2,3]
Olalekan L Aiyegbusi,[3,5,6,7] Derek Kyte [3,8]

For numbered affiliations see end of article.

**Correspondence to**
Mrs Nicola Elzabeth Anderson;
NEA451@bham.ac.uk

**ABSTRACT**

**Objectives** Patients undergoing haemodialysis report elevated symptoms and reduced health-related quality of life, and often prioritise improvements in psychosocial well-being over long-term survival. Systematic collection and use of patient-reported outcomes (PROs) may help support tailored healthcare and improve outcomes. This study investigates the methodological basis for routine PRO assessment, particularly using electronic formats (ePROs), to maximise the potential of PRO use, through exploration of the experiences, views and perceptions of patients and healthcare professionals (HCPs) on implementation and use of PROs in haemodialysis settings.

**Study design** Qualitative study.

**Setting and participants** Semistructured interviews with 22 patients undergoing haemodialysis, and 17 HCPs in the UK.

**Analytical approach** Transcripts were analysed deductively using the Consolidated Framework for Implementation Research (CFIR) and inductively using thematic analysis.

**Results** For effective implementation, the potential value of PROs needs to be demonstrated empirically to stakeholders. Any intervention must remain flexible enough for individual and aggregate use, measuring outcomes that matter to patients and clinicians, while maintaining operational simplicity. Any implementation must sit within a wider framework of education and support for both patients and clinicians who demonstrate varying previous experience of using PROs and often confuse related concepts. Implementation plans must recognise the multidimensionality of end-stage kidney disease and treatment by haemodialysis, while acknowledging the associated challenges of delivering care in a highly specialised environment. To support implementation, careful consideration needs to be given to barriers and facilitators including effective leadership, the role of champions, effective launch and ongoing evaluation.

**Conclusions** Using the CFIR to explore the experiences, views and perceptions of key stakeholders, this study identified key factors at organisational and individual levels which could assist effective implementation of ePROs in

### Strengths and limitations of this study

► Qualitative methods yield rich data with face-to-face semistructured interviews allowing the interviewer to monitor non-verbal communications and clarify ambiguous responses.

► This study involved explorations of a prospective intervention, meaning some participants were unfamiliar with key concepts. Preinterview materials were shared to support and inform discussion and participation.

► The role of the researchers was carefully considered to acknowledge and minimise bias associated with beliefs and values. Steps were taken to mitigate risks, including use of reflective diaries, participant checking and multiple researchers involved in the coding process.

► While purposive sampling methods led to a diverse sample of participants, it is acknowledged that the sample did not include non-English speakers or carers. Further research is required.

► Data were collected before the coronavirus pandemic. The healthcare delivery landscape in the UK has changed, and it is possible that some attitudes and beliefs particularly around digital data capture may have evolved.

haemodialysis settings. Further research will be required to evaluate subsequent ePRO interventions to demonstrate the impact and benefit to the dialysis community.

## INTRODUCTION

The prevalence of end-stage kidney disease (ESKD) requiring treatment with renal replacement therapies such as haemodialysis (HD) continues to rise worldwide.[1] Both underlying disease and treatment are associated with a high symptom burden and reduced health-related quality of life (HRQoL).[2 3] Historically, outcomes such as mortality and

dialysis adequacy based on biomedical parameters have been used to inform the management of dialysis services and individual care.[4] However, there is now a body of established research[5–10] demonstrating the importance of also capturing patient-reported outcomes (PROs). PRO measures capture data directly from patients on how they feel or function, without requiring interpretation from others, using standardised symptom and/or QoL questionnaires,[11] and are often in electronic format (ePROs).

However, while patient-reported experience measures (PREMs), which allow patients to self-report their experience of receiving healthcare, have been collected since 2017,[12] PROs are still not routinely and systematically collected to manage individual patient care in the UK: meaning that many patients and some members of the multidisciplinary team caring for them are inexperienced in PROs and related concepts.

Yet the international body of evidence exploring the use of PROs in nephrology settings is growing,[13–22] demonstrating acceptability and feasibility of ePRO capture[23–25] and how ePROs can support the delivery of patient-centred care.[26 27] However, the overall impact and benefit of PRO use in dialysis care is yet to be established.[28] In comparison, PRO research in oncology has shown defined benefits ranging from improved QoL and reduction in hospitalisations to overall survival[29] and even cost-effectiveness.[30]

In November 2020, an online UK Summit led by the UK Renal Association, entitled 'ePROs for the Kidney Patient Community', was held to create a comprehensive, UK-wide roadmap to facilitate and optimise the collection and use of ePROs for the benefit of people with chronic kidney disease (CKD). This summit highlighted the importance of key stakeholder engagement, including patients and front-line clinicians, early and at all stages of design and implementation.[31]

To inform and direct future research plans, this study aimed to investigate the methodological basis for routine PRO assessment through exploration of the experiences, views and perceptions of patients and healthcare professionals (HCPs) on implementation and use of PROs, particularly ePROs, in HD settings.

## METHODS
### Design
The qualitative research question: 'What are the experiences, views and perceptions of patients undergoing HD on the implementation and use of ePROs in routine care and research settings?'

Epistemologically, pragmatism, which is set within a paradigm of enquiry processes and research practicality,[32] provides the philosophical framework to answer this question. A core assumption of pragmatism is that research should proceed from a wish to produce actionable knowledge[33]; allowing the researcher to select the research design and the methodology deemed most appropriate.[34] These foundations led to the decision to use the Consolidated Framework for Implementation Research[35] to offer a theoretical perspective and a qualitative descriptive (QD) methodology.[36–38] This is particularly useful for healthcare studies focused on discovering the who, what and where of events or experiences and gaining understanding of inadequately understood phenomenon. Kim et al[38] describe QD as the 'label of choice' when an unambiguous description of a phenomenon is desired or information is sought to develop and refine questionnaires or interventions.[39] This choice of methodology led to the utilisation of the following methods.

### Participant selection
Participants included adults (≥18 years), receiving HD (in centre or at home), able to provide valid informed consent and converse in everyday English. Patients were excluded if they were not deemed established on HD or had been undergoing HD <3 months, or if they had an active intercurrent medical problem requiring enhanced routine clinical care.[40] Participants were identified and recruited between July 2018 and November 2019 by the lead author. Eligible patients were primarily approached face to face in the dialysis unit; patients dialysing in centre were approached before dialysis session and patients who dialysed at home (home haemodialysis (HHD)) were approached when they attended for clinic. Purposive sampling was undertaken to achieve maximum variation across age, gender, ethnicity, time on dialysis and comorbidities.[41] HCPs were recruited from the broader renal team, and initially contacted by email by the lead author; they included healthcare and administrative assistants.[40]

### Setting
Data were collected from 22 patients undergoing HD and 17 HCPs in the UK (see tables 1 and 2). All participants were being treated via, or working at, a large regional hospital. To accurately reflect the diversity within the dialysis population, patients being treated in centre (n=15) were recruited from two satellite units as well as seven patients choosing to dialyse at home. Non-medical members of the multidisciplinary team were targeted from these two units, one in a city and one serving mainly rural communities. All HCPs currently working in the home setting (n=5) had extensive previous experience of in-centre dialysis delivery.

### Data collection
All consenting participants took part in an audio-recorded semistructured interview with the lead author, either face to face or by telephone. Ethical approval was gained for follow-up interviews, but none were required. Patient interviews were conducted at the dialysis unit for those dialysing in centre (n=15) and at home for the HHD group (n=7). Since these interviews were conducted in patient homes, family members were sometimes present at the request of the patient, but did not take an active role in the interview. HCPs were interviewed in a private room at their workplace. Topic guides were used to steer

**Table 1** Patient participant characteristics

| Variable | Count (%) |
| --- | --- |
| Age | |
| <40 | 2/22 (9) |
| 40–49 | 3/22 (13) |
| 50–59 | 5/22 (23) |
| 60–69 | 3/22 (14) |
| 70–79 | 6/22 (27) |
| >80 | 3/22 (13) |
| Gender | |
| Male | 12/22 (55) |
| Female | 10/22 (45) |
| Ethnicity | |
| White British | 16/22 (73) |
| Black British | 4/22 (18) |
| Asian/Asian British | 2/22 (9) |
| In-centre dialysis | 15/22 (68) |
| Morning sessions | 12/15 (80) |
| Home HD | 7/22 (32) |
| Current vascular access | |
| Arteriovenous fistula | 19/22 (86) |
| Arteriovenous graft | 0/22 (0) |
| Central venous catheter | 3/22 (14) |
| Charlson Comorbidity Score | |
| <2 | 1 (4) |
| 3–4 | 5 (23) |
| 5–6 | 4 (18) |
| 7–8 | 6 (27) |
| 9–10 | 5 (23) |
| 11–12 | 1 (4) |
| 13+ | 0 (0) |
| Time since dialysis commencement (years) | |
| ≤5 | 11/22 (50) |
| 6–10 | 5/22 (23) |
| 11–15 | 3/22 (14) |
| 16–20 | 1/22 (4.5) |
| >20 | 2/22 (9) |

Charlson Comorbidity Index (CCI)[65] quantifies an individual's burden of disease and corresponding 1-year mortality risk. The index adjusts for 17 comorbidities, each one classified with a validated score of 1–6 points, based on the adjusted relative risk of 1-year mortality. The final total score is used to calculate the probability of survival. The index is being used in this context to demonstrate the overall disease burden of the population under study.
n=22.
HD, haemodialysis.

**Table 2** Healthcare professional participant characteristics

| Variable | Count (%) |
| --- | --- |
| Gender | |
| Male | 5/17 (29) |
| Female | 12/17 (71) |
| Ethnicity (Office of National Statistics categories) | |
| White British | 9/17 (53) |
| Other white background | 2/17 (12) |
| Black British | 1/17 (6) |
| Asian/Asian British | 1/17 (6) |
| Any other Asian background | 2/17 (12) |
| Black African | 1/17 (6) |
| White and Black Caribbean | 1/17 (6) |
| Role | |
| Consultant nephrologist | 5/17 (29) |
| Consultant surgeon | 1/17 (6) |
| Registered nurse | 9/17 (53) |
| Non-registered healthcare assistant | 1/17 (6) |
| Administrative assistant | 1/17 (6) |
| Time working across HD setting (years) | |
| ≤5 | 1/17 (6) |
| 6–10 | 4/17 (24) |
| 11–15 | 3/17 (18) |
| 16–20 | 2/17 (12) |
| >20 | 7/17 (41) |

n=17.
HD, haemodialysis.

and support the interview process (see online supplemental files 1 and 2). These guides were piloted with one participant from each group and then refined iteratively during the collection phase in response to initial findings. Field notes and in-depth memos were created after each interaction. Since PROs are not routinely collected in the UK, including the hospital trust where this research was undertaken, it was recognised preinterview information was required to aid the quality of discussion around PROs and future implementation. Therefore, participants were provided with a diagram of a core outcome set selected specifically for use in HD trials, to illustrate the position of PROs in outcome measurement[42] and example PROs from a recent systematic review and supported by our local renal public and patient involvement group: Kidney Disease Quality of Life Short Form,[43] Kidney Disease Quality of Life 36[44] and Integrated Patient Outcome Scale-Renal.[45] Patients in all UK renal units are invited to complete the annual renal PREM to report their experience of kidney care. This report demonstrates variation in experience across centres and can be used to drive organisational change and improvement[12]; some patient

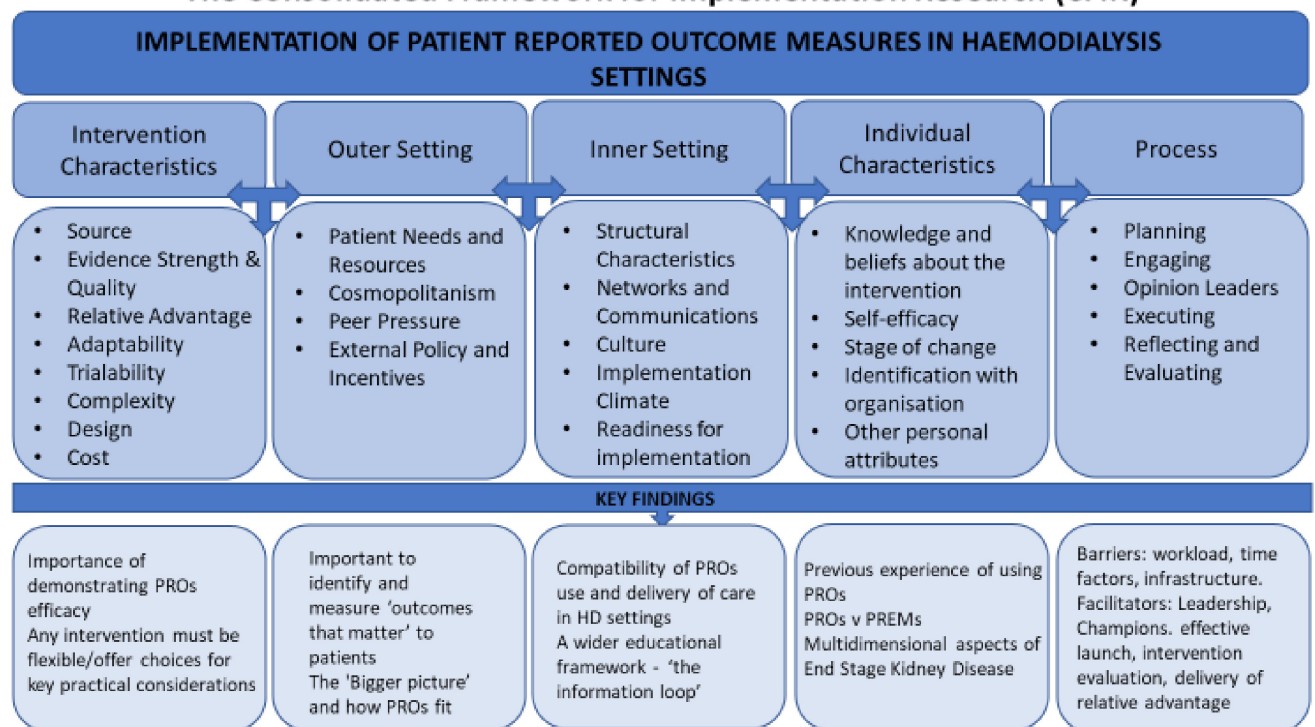

1.      Adapted from Damschroder LJ, Aron DC, Keith RE, Kirsh SR, Alexander JA, Lowery JC. Fostering implementation of health services research findings into practice: a consolidated framework for advancing implementation science. Implement Sci. 2009;4:50.

**Figure 1**    Conceptual framework and key findings (adapted from Damschroder *et al*[35]). HD, haemodialysis; PREM, patient-reported experience measure; PRO, patient-reported outcome.

participants indicated they had completed the annual PREM.

## Analysis

Using codebook thematic analysis,[46] a coding framework, drawing on the Consolidated Framework for Implementation Research (CFIR),[35] was used to deductively analyse transcripts, with further new subthemes developed inductively through data engagement and the analytical process.

The CFIR is a widely used conceptual framework developed to guide systematic assessment of factors that might influence implementation and effectiveness, including assessing potential barriers and facilitators in preparation for implementing an innovation (see figure 1).[35 47] Primary data analysis was conducted by the lead author, with two investigators (CMcM, DK) reviewing coding for consistency and appropriateness. As the lead author was a renal research nurse conducting this research as part of a Clinical Doctoral Research Fellowship, she was known to some HCP participants. This information was declared and discussed during the valid informed consent process. Additionally, a reflective research diary, memo writing and discussion with the study management team were used to try to minimise the influence of prior relationships on analysis. Data analysis was supported by qualitative data analysis software—QSR NVivo V.12. Participant verification,[48] to check that the transcript correctly documented

the discussion, was undertaken on one transcript from each group—no discrepant comments were reported. Data collection and analysis continued until saturation was achieved, that is, no new information pertinent to the research question was being generated by further interviews (see online supplemental tables 1 and 2).[49]

## Patient and public involvement

Research on the use and implementation of PROs in nephrology settings was prioritised by the local kidney patients charity. An existing patient and public involvement (PPI) group was used and a study-specific PPI group was convened to help develop the research question, design this study and the associated fellowship application. Patients were consulted on the study documentation, including topic guides and example materials (see online supplemental files 1 and 2). A summary of study findings will be made available for study participants.

## RESULTS
### Participant characteristics

Participant characteristics are summarised in tables 1 and 2.

The patient sample was broadly representative of prevalent HD population in the UK.[50]

Data saturation was deemed achieved after interviewing 16 HCPs and 15 patients (see online supplemental tables 1 and 2 for further detail).

## Key findings

Analysis identified the following themes, which are presented in line with the CFIR key domains. Although presented in a linear fashion, the five domains and their respective constructs cannot be considered in isolation, all interact to effect implementation.

Figure 1 shows the conceptual framework and associated findings. Illustrative quotations are provided in table 3.

### Intervention characteristics: factors associated with design and quality of the intervention, including how intervention is perceived
*The importance of demonstrating PRO efficacy and impact*

Both patients and HCP participants felt that PROs could support the delivery of person-centred care through shared decision-making and management in dialysis settings. However, HCP interviewees highlighted the key challenge was not the physical collection of PROs, but how the data were then used to improve outcomes. For patients, it was key that any time spent completing questionnaires should be rewarded with review and appropriate action, that is, not a 'tick-box' exercise.

HCPs, particularly nephrologists, extolled the importance of PROs in research settings.

### Intervention flexibility

The topic guide initiated discussions with participants on key practical considerations, including frequency (of PRO completion), optimal timing (around dialysis), preferred setting (home or in centre), favoured mode of administration (electronic or paper versions) and interpretation and feedback of the responses.

It was clear from the patients that a fixed means of implementing PROs would not meet the requirements of this heterogeneous group. Any system would need to be flexible while maintaining maximum simplicity. Most patients felt they could complete electronic measures if required, but some strongly favoured paper options. Most interviewees stated that self-completion was possible but felt some would need physical or emotional support from carers. A few patients indicated a desire for assistance with information technology (IT) aspects until they became familiar. Several patients alluded to the need to 'compartmentalise' their dialysis by conducting all dialysis-related activity in centre or by having discrete facilities and times for dialysis if being treated at home; suggesting that PRO completion would need to be undertaken at defined times/settings within their schedule. Others were less concerned about daily reminders of their disease and management, so were open to completion timings/settings that were more variable.

HCPs discussed the promise of electronic capture to support enhanced management of chronic symptoms, but some expressed concerns about potentially missing acute signs and effectiveness of associated safety reporting and actions. Two HHD patients foresaw the potential benefits of automated safety alerts associated with electronic capture. However, one patient noted that it was important that any automated information or self-help advice should not conflict with information given face to face by the doctor.

Participants were asked for their perspectives on computer adaptive technology, a type of assessment in which questions are generated specifically for each individual, using item response theory. This was a new concept to nearly all participants, but most quickly grasped the underlying theory after a short explanation and were supportive of the idea, recognising the potential to stop redundant questions and save time. (See table 4 for overview of practical considerations.)

### Outer setting: the wider societal, economic and organisational contexts in which the stakeholders and organisation implementing the intervention reside
*Identifying outcomes that matter*

There was a general agreement among all participants that it was important to ensure measurement of 'what outcomes matter most' to patients.[51] There was a broad agreement from patients that the example PROs they had been provided with were comprehensive in covering the key symptoms facing patients undertaking HD. Furthermore, they agreed with findings from the Standardised Outcomes in Nephrology (Haemodialysis) initiative (SONG-HD)[42] that fluctuations in HRQoL mattered more, to them, than biomedical outcomes. The importance of these outcomes was acknowledged by the HCPs, but it was felt that deep exploration of some PRO data fell outside the remit of the nephrologist or dialysis nurse and there was anxiety about acting outside their competency and the associated risks of litigation.

### The 'Bigger Picture'

Several patients highlighted problems with continuity of care and lack of cohesion between primary care and their dialysis provision. One HHD patient felt the use of PROs could assist uptake of home therapies by demonstrating better overall outcomes/HRQoL for patients managing their own dialysis. Participants in both groups reflected that PROs could help support shared decision-making by targeting and prioritising discussions according to the patient's agenda.

### Inner setting: the structural and cultural contexts around where the implementation will take place
*Compatibility of PRO use and approaches to care in HD settings*

Discussing potential implementation and how participants perceive the current situation as needing change, patients and staff alike discussed the approaches to care within dialysis settings. Patients clearly described a 'task orientated' style of care in centre, with a focus on practically administering HD, versus a more patient-centred model discussed by those dialysing at home. HHD patients dialysing themselves felt they had developed a level of expertise and the increased 'control' had improved their HRQoL. They indicated current communication pathways were effective and that the HHD team was responsive

**Table 3** Key findings—summary of direct quotes

| Key finding | Illustrative quotation | |
|---|---|---|
| | **Patient participants** | **HCP participants** |
| **1. Intervention characteristics** | | |
| The importance of demonstrating efficacy/benefit | 'If they haven't got a point for collecting them that isn't going to be actioned or put into a specific piece of research, then they shouldn't be there…. 'Yes, it makes you unwilling to invest your own time in something that you hadn't seen any benefit from in the past…. If it's just a box ticking exercise for your unit or whatever, to say that I've complied with some directive and nothing is going to change, then why would you.'.' Patient JI18 | 'one is convincing those patients and doctors and nurses that it's [PROs] actually of use to man or beast. I think that is going to be the main barrier.' HCP11 |
| The intervention must be flexible/offer choices | | 'I think if you could then see that it changed a consultation that meant that you went, 'Oh, I didn't know you were having that issue; I can do something about that'. The difficulty is if they then, it drives a lot of conversations that, you know, saying, 'Yeah, can't really do anything about then', and then everybody may well think well this hasn't changed anything so, you know, is it making you feel better writing it down? Don't know really. So if there's not an outcome difference to it then it just becomes a kind of form-filling exercise but there's not obviously a benefit.' HCP16 |
| Practical considerations | 'Yeah, but certainly filling it in is not an issue, I think discussing it and whether somebody has their partner with them or carer or whatever, I think it's something you should offer them and it's for them to make up their mind.' Patient MN07 | |
| | See table 4 for key findings and illustrative direct quotes. | 'I guess we have to give the option to the patient whether and how they want to fill this in or not because some patients may not want to fill it in. And I can think of a few patients on my dialysis unit that probably would not want to fill this in.' HCP14 |
| **2. Outer setting** | | |
| Identifying and measuring outcomes that matter | 'I need to go out. I need to keep this going. I need to keep feeling that I'm worth something by bringing home a wage and working.' Patient YZ13 | 'Yes, definitely. We see that all the time don't we? I mean you know, we see that all the time. I mean that's the battle, to get people to dialyse three times a week rather than twice a week [yeah, yeah]. So you know, I'm having this battle with a man in XUnit at the moment who's decided to go and dialyse twice a week and he tried it for a few weeks. And he said, 'Oh, I feel perfectly well. So I'm just going to carry on dialysing twice a week' and I said, 'Well, that's not the point' [laughs] [absolutely]. And he said, 'Yeah, but I feel fine. I get more time, I get more spare time'.' HCP13 |
| How PROs fit into the 'Bigger Picture' | 'It does really knock the quality of life, some people tend to be more worried about the quantity of life but I'm interested in the quality of life because not having quality of life, I can understand why some people do commit suicide, there's no quality.' Patient ST10 | 'I think so, I think it [useful for GP's to be able to see the patients results, answers to some of these questions] would because most of this is in the community so it would be very, very useful for the GP's to be able to access all the information.' HCP04 |
| | Continuity of care and communication with primary care:<br>' 'Yeah, the other thing I find with my GP for example, now she rang me, I have monthly blood test here and every 12 months you go to the GP and I go to the diabetic clinic, my diabetes is good… so the doctor (GP) rang up and said, I think you're anaemic, I said well I've had these blood tests at the dialysis centre and nobody phoned up that I was anaemic….' Patient ST10 | |
| **3. Inner setting** | | |
| Compatibility of PROs and approaches to delivery of care in HD settings | Task-centred care versus patient-centred care:<br>'yes, you might be surrounded by them [unit staff] but they might not be asking questions, you might not feel you can ask questions. There might be aspects that you might not necessarily think it is worthwhile asking.' Patient JI18 | Task-centred care versus patient-centred care:<br>'we don't always see the bigger picture, we're so focused on getting that needle in to get dialysis, that we forget, oh my god, there's a person on the end of that arm….' HCP12 |
| | | 'in-centre, it's looking at the patient, home haemo it's looking at the person.' HCP08 |
| | | 'I think the nurses have got time to get to know the patients to a certain extent as far as medically, their medication is involved. You know, how poorly they are, obviously what they need. Unfortunately, I don't think they have time to carry that a bit further….' HCP02 |

Continued

**Table 3** Continued

| Key finding | Illustrative quotation |
|---|---|
| PROs as part of a wider educational framework—'the information loop' | Health literacy of patients: 'Talking about monthly QA reports — '…and we all get these, telling us about our potassium, and somebody says to me, do you understand them? I say, I haven't got a clue, he says nor me, I says, all I know, when I go in there, they tell me that's okay, that's okay, that's okay, I said but I haven't got a clue.' Patient BA14 |
| | Implementation must be part of a wider framework of training, education and support: 'this will be exciting if implemented correctly, but there's a whole massive amount of work, it's not just this, it's everything else around it yeah, I think so.' HCP17 |
| | Staff education requirements: 'So I guess education would be vital, but they're about communication skills, they're about learning what matters in chronic disease management. And it would also be about, because that's burdensome, it's hugely draining to do it day in day out with a similar group of people. And you would need to put in support, raising the education for them is one, raising awareness. But I also think if you are hearing people's symptoms, which are as I say some of them you can't resolve and you're hearing them day in day out, you need somewhere to offload to as well. So it's like, but building a good structure around what's needed. But that doesn't mean don't talk about them.' HCP05 |
| **4. Characteristics of individuals** | |
| Previous experience of using PROs | Have you ever completed anything like these questionnaires before? 'No I don't think so.' Patient UV11, Patient BA14 'No I haven't.' Patient WX12 'I think I have, but I can't remember, it must have been some time ago…I don't think there were as many questions as that one.' (KDQOL-SF) Patient DC15 |
| | 'I mean I've used them in various research projects…And that's one of the reasons why I haven't routinely used them in clinical practice because I think they're fine for research, but I haven't found one that's hugely clinically, that I feel I'd really want to use.' HCP06 |
| Concept of PROs versus PREMs | Interviewer: Have you been asked to complete anything like that [exemplar PROs] before? Patient MN20: I've done questionnaires before, I think. Interviewer: Have you? What kind of questions did they ask? Were they similar ones? Patient MN20: Most of them have been in here about the staff and stuff about dialysis and how long I've been on it. It was similar. | '[PROMs & PREMs] I get confused and I think there's crossover and I find it quite difficult to draw a line between experience and outcome. Because for example, the experience of a patient. So, tiredness for example can be strongly affected by a patient experience. So, you have travelled six hours to get there or waited for your transport, your experience, it affects your tiredness which is—I think—I'm not sure always it's a useful distinction.' HCP12 |
| Multidimensionality of ESKD and consequent challenges | '…obviously the doctors can't cure that, they can't cure that, they can only cure your medical condition, they can't cure your mental condition so that's for you, but they can make it as well, so they give you the positivity to carry on.' Patient WX12 | Interviewer: Yeah. So do you think we should be measuring them (PROMs and PREMs) at the same time? HCP16: I think that they should be part of the same thing. I think that they should be seen so that we can pick apart whether people are happy with the service that they're getting but also how that interplays with their health and their feelings and their symptoms so I think the two things overlap. |
| | '…it's most of the time that I get anxious with social situations because of my other medical problems. It's not just about my kidney problems and then dealing with the social aspect. It's my kidney problems, plus my other medical problems and then the social situation. I have to deal with my other medical problems before I can even contemplate going out with family or friends…or shopping and things like that.' Patient WX12 | 'If we can't do much about it, then it's not our problem, sort of thing. That's one of those barriers; I think there's the worry that it uncovers things that are uncomfortable to talk about….' HCP15 |
| | 'The main problem? [with dialysis] Social life. Absolutely. It's not just my social life; it's the wife's social life as well. She's like a nurse to me really. Without wife, I don't know where I'd be. I've got two lads but you can't rely on them to be here.' Patient YZ13 | 'Acknowledgement that there are aspects of care that do not fall within doctors 'remit' 'things that aren't necessarily medical, that may relate to social services or other things that are driving a symptom or feeling that relates more to social care, to a dietitian or other members of the team….' HCP16 |
| **5. Process of implementation** | |

Continued

**Table 3** Continued

| Key finding | Illustrative quotation | |
|---|---|---|
| Supporting the process of implementation | Importance of leadership and champions:<br>'Yeah, Yeah, I've helped a lot of people. When I used to go to look at their machines, because they send me first to look at the room and whether it can be done, they didn't even know I was on dialysis. They'd look at me and say 'you seem to know a lot about this'. I said, 'yeah, because I'm a dialysis patient'.' Patient YZ13 | Importance of leadership and champions:<br>'Time, resources, you know, if someone's not here, say, if it was me implementing it, if I'm not here, I'm thinking, will it get done? It's as simple as that.' HCP17<br><br>'You might need to bring somebody else in from outside, yeah but that runs its own complications so what's the answer, a patient representative might be an answer, somebody who knows dialysis.' HCP17 |
| Barriers to effective implementation: time/attitude/IT problems | 'They [Doctors] haven't got time anymore. We all know that, but it's nice to not be rushed along and ignored.' Patient MN20<br><br>'It does annoy me when we have problems with logging in and passwords.' Patient WX12 | Bottom-up approach:<br>'Often there are new things that are introduced and when you've just got to do it attitude. Now that just gets peoples backs up to start with…have a proper process in place to prepare people.' HCP07<br><br>'So it has to be easy access for the patient. It's pointless having something they can't actually get to and that's scary or frustrating to get onto, to log on or something, yeah.' HCP05 |
| Facilitators to effective implementation: how PROs might deliver a relative advantage over current systems of care delivery | Acts as a communication tool:<br>'Yeah, yeah, you know, if you had a form to fill in, you're divorced from it because obviously within our human nature we always, we don't want to look vulnerable, we don't want to look vulnerable, right, so therefore, if the doctors say, you might be feeling sick, and you think, well I only get it now and again, to yourself, this is, so you don't mention it but if you mentioned it, he or she can make a diagnosis on that,….' Patient BA14<br><br>'I couldn't seem to settle in, i.e. couldn't communicate with the nurses, me as a person, I don't communicate very well.' Patient QR09<br><br>Help target the consultation: save time.<br>'Probably save your doctor from doing some of the work as well…As I say, if I'm not giving information that I need, we're not getting anywhere, are we?' Patient AB01<br><br>Help deliver person-centred care.<br>'If we've got monthly bloods and we've got the information on that at the same time, they can look at that [ePRO] instead of looking at the computer and seeing what's really going on with that person….' Patient AB01<br><br>'Even if that doctor is reading that one-on-one, he can look at this…even if it's on a computer, he can look at it and say, 'He's got a problem with that area. Let's go and deal with that'.' Patient AB01<br><br>'These questionnaires are good as well because a couple of symptoms cropped up that I do get but that I forgot is related to everything, I thought, 'Oh, that's why I get that then. Oh, that's why I get that'.' Patient WX12 | Could reduce workload:<br>'I find it really interesting how people would say that this is an increase in workload, when actually this is things that you need to be looking at anyway. For me this is nothing out of what I'm already looking at in every patient and of course it will be an increase in workload if you don't look at them.' HCP17<br><br>Potentially time saving:<br>'I think the PROMs, certainly one is useful when they fill them in but quite often they haven't filled them in and it's then sometimes difficult to ascertain have they not filled them in because they have got symptoms but they just aren't necessarily engaging or thinking oh therefore I will fill this in or whether they haven't filled it in 'cause they're fine. But then at the same time it's quite easy to ask a patient and say, 'Have you not filled this in?', because actually they're fine and then they'll say, 'Yes, I'm fine. There's nothing to worry about', sort of thing and then in a way you can think well I can, I don't necessarily need to give you targeted questions 'cause you've already got the opportunity to do that. Sometimes it then speeds things up.' HCP06 |

ESKD, end-stage kidney disease; GP, general practitioner; HCP, healthcare professional; HD, haemodialysis; IT, information technology; KDQOL-SF, Kidney Disease Quality of Life Short Form; PREM, patient-reported experience measure; PRO, patient-reported outcome; PROM, patient-reported outcome measure.

**Table 4** Practical considerations and illustrative direct quotes

| Practical considerations | Key findings | |
|---|---|---|
| | **Patient participants** | **HCP participants** |
| Frequency (to allow for accurate recall) | Wide variation of responses from 1 week to 12 months. Agreement that it needed to match recall while allowing for treatment responses. Several patients would complete as frequently as asked—demonstrating no preference. | Wide variation of responses from, as required to 12 months. Focus on practicality, that is, how often they thought patients would complete and time burden rather than sensitivity to changes and recall. Before routine clinic visit was a popular response. |
| Timing (in relation to dialysis cycle) | Some patients wanted to complete during dialysis but noted this could be practically difficult, particularly if their access was in dominant arm. Most did not want to complete immediately after dialysis, the majority requiring some recovery time. | *'Horses for courses' HCP11* meaning that people are different and will need different things. In terms of timing, some will need a short amount of time to complete, others will need longer. Options, in centre before or during session or patient's choice at home (need time frame for completion? ie, within 5-day window). General agreement that straight after dialysis patients are very fatigued, so not optimum time for completion. Effects of cognitive status on concentration levels—ability to partially complete and come back later, recall bias will be relevant. Need to consider are you asking about today, last 2 weeks, last month, when designing a tool. |
| Setting of completion (home/in centre) | Some favoured completion in centre, others preferred to complete at home, particularly if carer assistance required. | Patients needed options (home/unit)—some patients need assistance so carers could help (unlikely nurses would have time). It was suggested better returns if encouraged to complete at unit: at home patients are 'distracted by life'. *'Will responses be different if completing at home—does that matter? 'either way, you're gonna get the reality of their life, so perhaps it doesn't matter'.'* HCP05 Lack of privacy in centre noted by number of respondents. |
| Mode of capture | Varied responses—digital, paper or no preference. Proponents of digital capture displayed no obvious preference for choice of device. Only one patient had no access to a digital device at home. Preference for paper often reflected lack of confidence in ability to complete digitally, a number said they would need assistance to complete electronically. *'I feel like my confidence is down and that needs to build back up again before I use the computer.'* Patient CD02 | Advantages of electronic collection raised (real-time review, less admin, less likely to lose returns, easier to track responses, to review responses/feedback) but option of paper needs to be available, as some patients would be unable to complete digitally, and who would provide this assistance? Some pondered that current digital resources, that is, PatientView, are not widely used. IT systems require resource for infrastructure, upkeep and training. |
| Readability | No jargon or medical terms—got to be easy to read and understand. Guidance around more complex constructs. Availability of measure in other languages. *'You may find you need to print them in their own Urdu or whatever, so that they do understand them. And that has to be very careful that the question doesn't get lost in the translation.'* Patient MN07 | Simple and clear rating scales for staff undertaking interpretation. Need to be culturally sensitive/valid and available in languages other than English. |

Continued

**Table 4** Continued

| Practical considerations | Key findings | |
|---|---|---|
| | **Patient participants** | **HCP participants** |
| Length of questionnaire | Generally, length was not an issue, provided patients could see a purpose to completion. There was a dislike of repetitive questions. Potential time burden of lengthy questionnaires, seen as an inconvenience for carers. *'So the other question I was going to ask was were you put off by the length of some of those questionnaires?'* Interviewee: *'No I wasn't, no, you know I'd put aside the time to do it so no I wasn't, I could see the point of them.'* Patient UV11 | Consensus that the shorter the questionnaire, the better. Some commented that current validated PROs were much too lengthy. |
| Form of feedback | No overall preference: could be letter, email or discussion. Most happy to use graphical data. However, it is important that timely feedback given and PRO completion *'not just a tick box exercise'*. Some patients wanted carers involved in feedback process. *'Because I don't do emails and sometimes my eyesight, I can't able to read. Yeah. I prefer in person.'* Patient GHO4 <br><br> The need for timely feedback was made: <br> Interviewer: *So you'd want to see some sort of feedback within two weeks.* <br> Interviewee: *'Two weeks and they sort it out. Not leave it for months and months and years and years, no good at all.'* Patient KL06 <br><br> *'As well as giving helpful information to help you manage your condition, it would also be useful to have information about what causes certain things in order to put your mind at rest, if you know what I mean?'* Patient CD02 | Consensus that electronic feedback is ideal. Responses outlined in letter to patient or face to face in clinic to allow 'probing'—but this requires resource. Agreement that longitudinal data would be advantageous to see trends—preference for graphical rather than tabular form—but scales need to be well thought through, 0–3 not considered 'sensitive' enough to be able to show changes—preference for EQ-5D thermometer scale… *'but a 'Blunt tool'— good clinician should pick up sensitive changes before PROs would.'* HCP11 No time to read lots of free text. |
| Setting of feedback | Currently most consultations take place while the patient is dialysing. Patients gave mixed responses to the wish to be seen in private, most did not want additional or extended visits at the dialysis unit and seemed resigned to being seen in open bays. | If medical staff were providing feedback in person, pragmatically best to do this in clinic, it was acknowledged this feedback can take place at the bedside during dialysis but participants noted this may affect the quality and content of the discussion. |
| Who provides feedback | No overall preference, but one patient felt more comfortable talking to nursing staff. | Variation on responses—some thought medical staff (but concerns raised around time and workload), others thought nurses (some anxiety regarding time constraints and knowledge/experience of junior nursing staff to know how to manage symptoms or issues raised by PROs). *'Need clear process for review and feedback—' 'can't have 6 nurses going back to patient to discuss responses'.'* HCP16 |

EQ-5D, EuroQol-5 Dimension; HCP, healthcare professional; IT, information technology; PRO, patient-related outcome.

and holistic in their approach to care. Analysis of patient and HCP data suggested that time constraints were the major reason for a task-based approach in centre with both patients and staff exhibiting a desire to complete dialysis sessions with as little impediment as possible. Correspondingly, most in-centre patients did not feel staff had time to have long discussions and some suggested PROs might help rectify this. There was no discernible variation in the views of the HHD cohort on the practicalities of PRO collection.

### Part of a wider framework: the 'information loop', the role of PROs in addressing training needs and health literacy

HCP data analysis highlighted the importance of education, training and support for successful implementation. It was indicated that while medical staff needed training in PRO interpretation, such as rating scales, nurses might need a broader programme on use of PROs as well as associated chronic disease management. There was disparity indicated in the health literacy of the patient group; some HHD patients appeared expert in their illness and treatment, while some in-centre patients appeared less confident in their knowledge.

### Characteristics of individuals: factors associated with individuals involved in implementation
#### Previous experience of using PROs

Patients expressed an overall lack of awareness and experience of PROs. Many had completed annual PREMs but highlighted a lack of feedback or action. Some patients expressed general anxiety about questionnaire completion, linked to their experience of externally administered questionnaires for other agencies: for example, Personal Independence Payment surveys assessing entitlement to extra costs associated with long-term ill health or disability. When reviewing the example PROs, it was noted that some patients, particularly those from minority ethnic backgrounds, flagged the potential difficulty around asking about sensitive issues such as sexual function. While the participants themselves were happy to discuss such issues, they recognised that not all patients might be.

All nephrologists interviewed had experience of using PROs in research settings, but none had regularly and systematically collected PROs in routine care. PRO use in any setting was largely a new concept for nursing staff.

There was a confusion around the concepts of PROs versus PREMs, expressed by both patients and HCPs. Participants found it hard to distinguish experience of care from outcomes and used terms interchangeably.

### Multidimensional aspects of ESKD and consequent challenges

Many of the patients in the study had multiple comorbidities and associated symptoms, reflective of the wider HD population. Patients indicated that it was often hard to know which symptoms related to their ESKD and/or treatment, and which were associated with other diseases or advancing age; leading to anxiety about what to raise

during a consultation. It was suggested that PRO data review and feedback gave a chance to discuss symptoms and potential causes, thereby potentially providing reassurance. However, some clinicians thought it was outside their role to manage non-nephrology-related outcomes presented by PROs and that it would be difficult to disentangle several, possibly unrelated, symptoms. They feared focusing on symptoms that were not currently being experienced or intractable could cause frustration or anxiety for themselves and the patients. Some patients gave a contrasting opinion, revealing the chance to be heard, even without a solution, was often sufficient to maintain the patient–clinician relationship. There was also concern that an over-reliance on PROs could result in a distraction from other important clinical issues.

The experience of life on dialysis was highly varied, with some participants exhibiting signs of depression, anger or acceptance, as well as reporting multiple symptoms.

### The process of implementation
#### Supporting the process of implementation

The analysis identified leadership, the identification of champions and a 'bottom up' approach to communication and shared solutions among both patients and clinicians as key factors supporting implementation. All participants emphasised the importance of support during completion; however, patients highlighted this need not always be delivered by healthcare staff. Peer-to-peer support and non-clinical champions, that is, administrative staff, might assist. HCPs felt a comprehensive launch was important and that individual roles should be clear. They reflected that senior members of the clinical team would be opinion leaders, and any change agents would need to understand the dialysis setting. Evaluation and reflection were important process components to recognise and deal with any unintended consequences.

### Overarching themes: barriers and facilitators to effective implementation

Across all CFIR constructs, analysis identified potential barriers and facilitators to the introduction of PROs.

### Potential barriers

Nephrologists cited a lack of evidence base supporting the use of PROs in routine kidney care and were concerned regarding the risk of overmedicalising the patient experience. There were perceived time barriers for staff, that is, workflow interruptions, additional obligations caused by PROs; with nurses perceiving quality time with patients in centre as limited. HCP participants argued that nursing documentation had lessened available time, as it often required recording on digital devices away from the bedside. They feared patients were being overburdened by questionnaires, particularly the less health literate.

In contrast, patients were often already aware of the complications associated with ESKD and HD. Many had achieved a degree of acceptance and some were reassured that the symptoms were expected and not something new

to deal with. Frustration would only arise if, having taken time to complete the questionnaires, no action was taken. Both groups agreed that IT issues, that is, inability to log in, no Wi-Fi, could be a barrier.

### Potential facilitators

Several HCPs considered how PROs might work to deliver a relative advantage over current systems to support more patient-centred care. Participants felt PROs could be a communication tool—acting like a 'tin opener'; but requiring that responses be carefully probed, highlighting that this required appropriate skills and training. Others saw the measures as an aide-mémoire which could help target the consultation and even save time; as could the possibility of remote management of care using ePROs. Perhaps, most importantly, PROs were seen by both patients and HCPs as a way to get to know the patients and hence deliver more meaningful care.

### DISCUSSION

There is a growing body of published, peer-reviewed literature exploring the experiences of both kidney patients and multidisciplinary kidney HCPs with ePROs.[13 17 23 26 27 52] Previous studies have included non-dialysis-dependent CKD and peritoneal dialysis populations,[17 23 27] evaluating existing ePROs and associated delivery systems. This paper adds to the corpus by using semistructured interviews to gain rich data to inform and optimise future ePRO implementation in an HD population naïve to ePRO collection and clinicians unused to routine and systematic ePRO use.

The key findings of this study are that while patients and HCPs support the concept of PROs, further evidence of their potential benefit is required for effective implementation. It emphasises that any system should be flexible and measure what matters most to patients. Most importantly, the data collected should be acted on. There was a general lack of awareness and experience of PROs particularly among patients and nurses, with concern among the HCPs that PRO capture may highlight issues they might have neither the experience nor resources to manage. Therefore, a comprehensive implementation strategy is required to support any delivery, which involves strong leadership, patient and clinician involvement and ongoing training.

This study highlighted the importance of getting 'buy in', that is, gaining acceptability, from these stakeholders. This could be achieved by demonstrating evidence on their potential benefits thereby increasing trust to warrant practice change. Clinicians particularly questioned whether positive effects on survival, reported QoL, patient–clinician communications and cost efficiencies demonstrated in oncology[29 53–55] could be replicated in HD settings.

The body of evidence around ePROs in nephrology is emerging. Studies from North America demonstrate the feasibility of electronic capture of PROs in HD[19 24] and an Australian pilot study is currently exploring the feasibility and acceptability of ePRO capture and feedback among patients receiving HD in the Symptom Monitoring With Feedback Trial.[20] In advanced CKD populations, the Renal Electronic Patient Reported Outcome Measures (RePROM) study is piloting the use of an ePRO for remote symptom monitoring in the UK,[56] while the Ambu-Flex telePRO system is used to manage renal follow-up in Denmark.[25] Aiyegbusi et al[17] explored key stakeholder perspectives on the use of PROs in these predialysis patients and their findings are consistent with this study, particularly regarding potential benefits and administrative aspects; suggesting PRO data collection early in the patient pathway can be instituted and continue as illness advances through to renal replacement therapy or even conservative care.

A review and synthesis of evidence is currently being undertaken to investigate how PROs might work to enhance patient-centred care in renal settings, to offer strategy and guidance at individual and aggregate levels of decision-making.[57] For effective multiple uses of data, implementation needs to be viewed within the context of complex data linkage and accessibility issues. Studies in the UK and Australia are currently exploring and testing such linkage of symptoms and QoL data to the Australia and New Zealand Dialysis and Transplant Registry and UK Renal Registry.[20 22 58]

However, the purpose of data collection must be clear. Participants in this study confused the concepts of PROs versus PREMs, understandable given that experience of undertaking HD is intricately linked to both HRQoL and symptoms. The integrated use of both measures, side by side, warrants further investigation. Anxieties about purpose and data sharing could affect patient engagement; clear communication of roles and expectations should be undertaken.

While guidance exists on what outcomes to measure in research[5 6] and routine practice[9] this has not necessarily been centred on routine capture and feedback to guide individual care. Consequently, it is not clear which measure(s) should be used to capture data in a non-burdensome way, while providing sufficient measurement properties to support both single patient monitoring and aggregation of data where required, meaning further enquiry is needed.

The general lack of awareness and experience of PROs and concern that PRO capture may highlight issues that HCPs should not or cannot deal with, leads to a significant finding that PRO implementation must sit within a broader educational framework. PROs could be used to support wider initiatives and training, especially in the nursing group, who can then cascade information and self-management skills to patients, thereby increasing overall patient health literacy. This could be considered as closing an 'information loop'. There is already evidence that patient education is associated with better patient outcomes[59] and new strategies and quality improvement programmes exist such as Shared HD, a UK programme

aiming to support patients receiving in-centre HD to be more independent and confident in participating in aspects of their own care.[60] Potential barriers are cost and resources, but increasingly, accredited online learning facilities are available. However, it is acknowledged, staff in this study indicated a preference for face-to-face training. Education and training on PROs would be needed for all stakeholders.

This study identified time and workflow interruptions as key barriers to implementation. Rotenstein et al[61] found that following initiation of routine PRO collection in surgical settings, such concerns shifted as clinicians became comfortable with new processes. There was even a suggestion that PROs could enhance physician satisfaction and prevent 'burnout'.[61] Tong et al[52] interviewed the nephrologists, who also identified resource constraints and uncertainties in how to prioritise, measure and manage a range of competing comorbidities and broader QoL outcomes in a clinical setting that is technically demanding and traditionally focused on biochemical factors; findings mirrored by this study. Such anxieties will need addressing.

During the SARS-CoV-2 pandemic, data from the UK Renal Registry showed that patients undergoing HD experience a relative risk of mortality of 45.5% compared with the general population.[62] This vulnerable population has been unable to effectively shield, requiring regular dialysis treatment with associated risks around shared transport and waiting areas. Various strategies have been employed to reduce risks[63] but new ways of care delivery were required. The pandemic has irrevocably changed healthcare, with increased use of virtual services[64] and remote clinic visits. The addition of ePROs to readily available biomedical data means that clinicians could more effectively deliver patient-centred care without the necessity of the patient being physically present. This study also suggests that remote symptom monitoring could offer patients the safety and confidence to dialyse at home and arguably ePROs could assist the HHD service in maintaining its patient-centred approach while serving ever increasing numbers.

There are limitations to this study. Participants needed to be English speakers; this could affect transferability of findings. Views of carers were not specifically sought but their influence and importance were clearly identified by patients, suggesting further research targeting this group is warranted. Critically, data collection took place before SARS-COV-2 pandemic and experiences and perceptions around digital data capture and new approaches to healthcare delivery may well have shifted.

To conclude, the SARs-COV-19 pandemic has caused an irreversible shift in healthcare delivery, with increased use of digital communication and assessment. While the nephrology community, both patients and HCPs, are largely supportive of the concept of ePROs, there remain caveats to their routine and systematic use. Stakeholders need to be convinced by empirical evidence, considering the best available measures and methodological considerations. By exploring the experiences, views and perceptions of major stakeholders, this study identified key factors at organisational and individual levels which would assist effective implementation of ePROs. Further research will then be required to evaluate any subsequent ePRO interventions to empirically demonstrate the impact and benefit of their use to the dialysis community.

**Author affiliations**
[1]Research and Development, University Hospitals Birmingham NHS Foundation Trust, Birmingham, UK
[2]Department of Renal Medicine, University Hospitals Birmingham NHS Foundation Trust, Birmingham, UK
[3]Centre for Patient Reported Outcomes Research, Institute of Applied Health Research, University of Birmingham, Birmingham, UK
[4]NIHR SMRC, University of Birmingham and University Hospitals Birmingham NHS Foundation Trust, Birmingham, UK
[5]NIHR Birmingham Biomedical Research Centre, University of Birmingham and University Hospitals Birmingham NHS Foundation Trust, Birmingham, UK
[6]Centre for Regulatory Science and Innovation, Birmingham Health Partners, Birmingham, UK
[7]NIHR Applied Research Collaboration West Midlands, University of Birmingham, Birmingham, UK
[8]School of Allied Health and Community, University of Worcester, Worcester, UK

**Acknowledgements** PROM-HD PPI Group, Renal Research Patient Advisory Group, Queen Elizabeth Hospital Kidney Patients Association (QEKPA).

**Contributors** NEA, CMcM, MC, MD, PC, OLA, DK: substantial contribution to the conception and design of the study. NEA: data collection. NEA, CMcM, DK: analysis and interpretation of data for the work. NEA, CMcM, MC, MD, PC, OLA, DK: drafting and revising the work critically for important intellectual content and final approval of version to be published and agreement to be accountable for all aspects of the work in ensuring that questions related to the accuracy or integrity of any part of the work are appropriately investigated and resolved.

**Funding** (Nicola Anderson, Clinical Doctoral Research Fellow, Grant Reference ICA-CDRF-2018-04-ST2-027) is funded by Health Education England (HEE) / National Institute for Health Research (NIHR) for this research project.

**Disclaimer** The views expressed are those of the author(s) and not necessarily those of the NHS, the NIHR or the Department of Health. The study sponsor and funders have no role in study design, including collection, management, analysis and interpretation of data; writing of the report and the decision to submit the report for publication.

**Competing interests** MC is an NIHR senior investigator and receives funding from the NIHR Birmingham Biomedical Research Centre, the NIHR Surgical Reconstruction and Microbiology Research Centre and NIHR Applied Research Collaboration West Midlands at the University of Birmingham and University Hospitals Birmingham NHS Foundation Trust, Health Data Research UK, Innovate UK (part of UK Research and Innovation), Macmillan Cancer Support, UCB Pharma and GSK. MC has received personal fees from Astellas, Takeda, Merck, Daiichi Sankyo, Glaukos, GSK and the Patient-Centered Outcomes Research Institute (PCORI) outside the submitted work. DK reports grants from Macmillan Cancer Support, Innovate UK, the NIHR, NIHR Birmingham Biomedical Research Centre and NIHR SRMRC at the University of Birmingham and University Hospitals Birmingham NHS Foundation Trust, and personal fees from Merck and GSK outside the submitted work. OLA is funded by the NIHR Birmingham Biomedical Research Centre and declares personal fees from Gilead Sciences and GSK.

**Patient consent for publication** Not required.

**Ethics approval** Ethics approval was issued on 13 September 2017 (Ref No: 17/WA/0281) by Wales 7 Ethics Committee.

**Provenance and peer review** Not commissioned; externally peer reviewed.

**Data availability statement** Data are available upon reasonable request. Deidentified data may be available upon reasonable request via the corresponding author (ORCID 0000-0002-0614-3198). Protocol available doi:10.1136/bmjopen-2018-021532.

**ORCID iDs**
Nicola Elizabeth Anderson http://orcid.org/0000-0002-0614-3198
Christel McMullan http://orcid.org/0000-0002-0878-1513
Melanie Calvert http://orcid.org/0000-0002-1856-837X
Derek Kyte http://orcid.org/0000-0002-7679-6741

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
