## [Reviewer comments · BMJ Open]

ARTICLE DETAILS

TITLE (PROVISIONAL)	Using Patient Reported Outcome Measures during the management of patients with end stage kidney disease requiring treatment with haemodialysis (PROM-HD): a qualitative study.
AUTHORS	Anderson, Nicola; McMullan, Christel; Calvert, Melanie; Dutton, Mary; Cockwell, Paul; Aiyegbusi, Olalekan; Kyte, Derek

VERSION 1 – REVIEW

REVIEWER	Kara Schick-Makaroff University of Alberta, Faculty of Nursing
REVIEW RETURNED	15-May-2021

GENERAL COMMENTS	Using patient reported outcome measures during the management of patients with established kidney disease requiring treatment with haemodialysis (PROM HD): a qualitative study Thank you for the opportunity to review your manuscript. The implementation and use of PROs in routine clinical dialysis care is a growing field. I believe that this paper will add to this body of literature, and will be of interest to the readers of BMJ Open. Overall The overarching question I could not quite figure out (until later in the Results) was whether contextually PROs were already completed, collected, and used in patient care in the context of this study, or if this was an exploration of future implementation and use in kidney care? Were participants naïve to PRO use and implementation? Some of this is addressed in results #4, but seemed to focus on PREMs, not PROs. Clarifying this earlier in the paper would provide clarity to the reader. Abstract/Introduction • In the aim statement of the abstract, you have “This study explored perceptions and experiences of patients receiving HD and HCPs on implementation and use of PROs in routine clinical practice.” You don’t appear to differentiate between perception and experience in the paper.o However, this aim statement should be the same as the aim statement in the paper. But in the paper, you state: “The aims of this project was to conduct a qualitative study to examine the acceptability and feasibility of routine PRO assessment in HD settings.” Either focus on perception of implementation and use of PROs or acceptability and feasibility, but these are not the same thing. The aim should be consistent throughout the paper and addressed directly in the findings. (As noted below, the summary of themes should address this aim / your research question.)o In the Data Collection section, you explain that PROs are
--

not routinely collected. Contextually, were they routinely collected in the unit/s where these patients received care? If not, is this work then to inform future implementation?

- Abstract: Is there a way to concisely state your thematic findings (e.g., in 1 sentence) in a way that they either address your aim or research question? This statement could be re-used at the beginning of your results / discussion.

- Abstract conclusion: I'm not sure that this statement fully reflects the "so what?" of your study (particularly given that facilitators and barriers were not the key focus of your study/analysis). This could be strengthened to provide the reader with your "final take-home" message from your study.

- Introduction: Please replace reference to use of PROs in oncology with what is known through studies in nephrology and/or dialysis - either evidence-based benefits for patient outcomes, or patient/HCP perceptions of use of PROs in clinical practice. (Also see my first statement below under Discussion.) The nephrology literature on this topic should be included in this section to show how your paper will add to the body of literature, and further address why this study needed to be done.

Methods

- Design

- o Pragmatism is typically viewed a philosophical approach, not a methodological approach to qualitative work (that I know of). So it makes sense that from a pragmatist orientation, your question guided the design towards a qualitative approach. Did you have a qualitative methodology guiding your study? IE qualitative description, descriptive exploratory, etc? (Braun & Clarke 2020 seem to locate TA as a method, not a methodology). Please add references.

- o What is your qualitative research question?

- Participant selection

- o Who approached patient participants? (E.G., a researcher, a third party?)

- o Did home HD and in-centre dialysis patients all come to this one location from which they were invited to participate? In the Results, it reads as if all patients dialyzed at home – so were they all on home HD? And were all HCP working with home dialysis patients? Pretty sure the answer is no give that in Table 1, we read that 15/22 were in-centre, 7/22 HHD – so please just edit paper so this is clear.

- The setting seems to be missing in the methods. Note that the statements in the first paragraph under Results about where participants were treated / worked, recruited etc seems to fit better under Methods than Results.

- o Who sent the email of invitation to HCP? (E.G., a manager, clerical staff, a researcher?)

- Tiny note that HCP is sometimes used, and other times spelled out in full throughout the paper (including in the abstract)

- Analysis

- o The use of CFIR is a strength of the study. Does your use of the CFIR assume that the "intervention" being discussed here is about potential use, actual use, or both potential and actual use and implementation of PROs in dialysis care?

- o What does it mean that "participant verification was undertaken on one transcript..." Perhaps clarify and add a reference?

	 • Patient Public Involvement – need to put (PPI) behind this to explain acronym Results  • Might Box 1 be adapted similarly to Box 2 so that both HCP and patient participant exemplar quotes were provided for each key finding and reflect the narrative description in the paper? • Box 2 – “Timing” row. I need help understanding that “horses for courses” means in relation to timing?  o Not sure what “IV” and “I” means before quotes o Box 3: A few grammatical mistakes / incomplete sentences; consistent use of quotation marks; use of “PROM” but paper uses “PRO” • Finding 3 – it seems contradictory that patients “described a task-oriented style of care...with a focus on practically administering HD...” and that the Home HD team was “responsive and holistic in their approach to care.” Or is it that patient quotes support both ends of this spectrum? • Finding 4 – a little of this context could be described in Methods/setting that PROs were not used routinely in care, but an annual PREMs were completed (for organizational use?). (This comes back to my earlier point that participants were naïve to PRO use and implementation, and in this study you are then addressing potential future use and implementation of PROs in dialysis care.) Discussion  • I would suggest that the first statement in the discussion is incorrect. There is a growing body of published, peer-reviewed literature that addresses use of PROs in dialysis care and the experiences of both kidney patients and multidisciplinary kidney HCP.  o For just a few examples, see PMID: 3519925; 32242823; 26591273; 31632681; 28651528; Smith V, Wise K. Ren Soc Australas J. 2017;13(1):14-21; doi:10.5750/ejpc.v4i2.1125 o This literature could then be considered to confirm your findings, or highlight what your study adds that has not previously been articulated. • Re. confusion between PROMs and PREMs – please clarify this statement to explain that PREMs are self-reports about individuals’ experiences of receiving healthcare -which is distinct from PROs – self-report of individuals’ health outcomes relevant to their QoL. Conclusion  • This could be strengthened to provide the reader with your “final take-home” message from your study. Supplementary files  • Table 1. # invited / # participating = 22/41 – but I’m wondering if these numbers are reversed and it should be 41 were invited and 22 participated? • Same comment for Table 2 re healthcare professionals • Tables 1 & 2 – do the black “x’s” mean that this was coded in NVIVO? (whereas grey “x’s” denote first coding in NVIVO) Thank you for the opportunity to provide feedback on your manuscript.
--	--

REVIEWER	Andrea Gibbons University of Winchester Department of Psychology
-----------------	---

REVIEW RETURNED	18-May-2021
GENERAL COMMENTS	This qualitative paper focuses on the use of PROMs during management of patients with kidney disease on HD. The topic is relevant, and is timely considering covid and the need for alternative approaches to patient management. The use of a pragmatic approach is appropriate, and there is considerable discussion of reflexivity, which is often lacking in qualitative papers such as this. My only comment would be that the interview questions are very closed and, in some cases, a little leading (e.g. should a 'good' clinician already be able to elicit those outcomes that are important to a patient?), which is very similar to the quote listed in Box 2 under Form of Feedback. Was the question added to the topic guide after this interview? Or is this quote in direct response to the question? I appreciate that the questions need to be focused, but I think it would be important to acknowledge the task-focused nature of the questions in the discussion, and to make clear the context of this and any other quotes that were in response to direct questions such as this.

VERSION 1 – AUTHOR RESPONSE

Response to Reviewer 1:

Many thanks for taking the time to review this paper. We greatly value the comprehensiveness of the review and guidance within your comments, which we hope have been fully addressed below.

1. Overall

The overarching question I could not quite figure out (until later in the Results) was whether contextually PROs were already completed, collected, and used in patient care in the context of this study, or if this was an exploration of future implementation and use in kidney care? Were participants naïve to PRO use and implementation? Some of this is addressed in results #4, but seemed to focus on PREMs, not PROs. Clarifying this earlier in the paper would provide clarity to the reader.

Response: Thank you for flagging this. This study explores the future use of PROs. This has now been clarified in the abstract, introduction and strengths and weaknesses sections. See tracked changes (**Abstract pages 2-3 lines 23-64, Strengths and weaknesses page 4 lines 81-84 , Introduction pages 8 lines 160-163 - all set out below**)

Pages 2-3 lines 23-64

'ABSTRACT

Objectives

Patients undergoing haemodialysis report elevated symptoms, reduced health-related quality of life, and often prioritise improvements in psychosocial wellbeing over long-term survival. Systematic collection and use of patient reported outcomes (PROs) may help support tailored healthcare and improve outcomes. This study investigates the methodological basis for routine PRO assessment, particularly using electronic formats (ePROs), to maximise the potential of PRO use, through exploration of the experiences, views and perceptions of patients and Healthcare Professionals (HCPs) on implementation and use of PROs in haemodialysis settings.

Study design

Qualitative study

Setting and participants

Semi structured interviews with 22 patients undergoing haemodialysis, and 17 HCPs in the United Kingdom.

Analytical Approach

Transcripts were analysed deductively using the Consolidated Framework for Implementation Research (CFIR) and inductively using Thematic Analysis.

Results

For effective implementation, the potential value of PROs needs to be demonstrated empirically to stakeholders. Any intervention must remain flexible enough for individual and aggregate use, measuring outcomes that matter to patients and clinicians, while maintaining operational simplicity

Any implementation must sit within a wider framework of education and support for both patients and clinicians who demonstrate varying previous experience of using PROs and often confuse related concepts. Implementation plans must recognise the multidimensionality of end stage kidney disease and treatment by haemodialysis, while acknowledging the associated challenges of delivering care in a highly specialised environment.

To support implementation, careful consideration needs to be given to barriers and facilitators including effective leadership, the role of champions, effective launch, and ongoing evaluation.

Conclusions

Using the CFIR to explore the experiences, views and perceptions of key stakeholders, this study identified key factors at organisational and individual levels which could assist effective implementation of ePROs in haemodialysis settings. Further research will be required to evaluate subsequent ePRO interventions to demonstrate impact and benefit to the dialysis community.'

Page 4 lines 81-84

Strengths and Weaknesses:

'This study involved explorations of a prospective intervention, meaning some participants were unfamiliar with key concepts. Pre-interview materials were shared to support and inform discussion and participation.'

Pages 8 lines 160-163

Introduction:

'To inform and direct future research plans, this study aimed to investigate the methodological basis for routine PROs assessment through exploration of the experiences, views and perceptions of patients and HCPs on implementation and use of PROs, particularly ePROs, in haemodialysis settings.'

2. Abstract/Introduction

In the aim statement of the abstract, you have "This study explored perceptions and experiences of patients receiving HD and HCPs on implementation and use of PROs in routine clinical practice." You don't appear to differentiate between perception and experience in the paper. However, this aim statement should be the same as the aim statement in the paper. But in the paper, you state: "The aims of this project was to conduct a qualitative study to examine the acceptability and feasibility of routine PRO assessment in HD settings." Either focus on perception of implementation and use of PROsor acceptability and feasibility, but these are not the same thing. The aim should be consistent throughout the paper and addressed directly in the findings. (As noted below, the summary of themes should address this aim / your research question.)

Response: Thank you, the 'Aim' is now consistent throughout paper. We have clarified that this study is an exploration for future implementation. This is now explicitly stated in abstract, introduction, methods and results sections. Whilst the focus was experiences, views and perceptions (understanding of experiences to date and views, perceptions of future collection and utilisation of PROs) this included discussions around concepts of feasibility and acceptability of ePROs to patients and health-care professionals, as well as potential barriers and facilitators.

Page 8 lines 160-163

'To inform and direct future research plans, this study aimed to investigate the methodological basis for routine PROs assessment through exploration of the experiences, views and perceptions of patients and HCPs on implementation and use of PROs, particularly ePROs, in haemodialysis settings.'

In the Data Collection section, you explain that PROs are not routinely collected. Contextually, were they routinely collected in the unit/s where these patients received care? If not, is this work then to inform future implementation?

Response: Yes, this study was being undertaken to inform future implementation as clarified in the research aim. See above responses and the methods: data collection section:

Page 11/lines 231-234

'Since PROs are not routinely collected in the UK, including the hospital trust where this research was undertaken, it was recognised pre-interview information was required to aid the quality of discussion around PROs and future implementation.'

Abstract: Is there a way to concisely state your thematic findings (e.g., in 1 sentence) in a way that they either address your aim or research question? This statement could be reused at the beginning of your results / discussion. Abstract conclusion: I'm not sure that this statement fully reflects the "so what?" of your study (particularly given that facilitators and barriers were not the key focus of your study/analysis). This could be strengthened to provide the reader with your "final take home" message from your study.

Response: The abstract has been rewritten to more clearly address the research question and strengthen the conclusion. See above and tracked changes on **page 3**.

Introduction: Please replace reference to use of PROs in oncology with what is known through studies in nephrology and/or dialysis - either evidence-based benefits for patient outcomes, or patient/HCP perceptions of use of PROs in clinical practice. (Also see my first statement below under Discussion.) The nephrology literature on this topic should be included in this section to show how your paper will add to the body of literature, and further address why this study needed to be done.

Response: The introduction/discussion has been rewritten to increase the focus on relevant nephrology literature. Please see tracked changes:

Page 7 lines 146-151

'Yet the international body of evidence exploring the use of PROs in nephrology settings is growing (14-23), demonstrating acceptability and feasibility of ePROs capture (24-26) and how ePROs can support the delivery of patient centred-care (27, 28). However, the overall impact and benefit of PROs use in dialysis care is yet to be established (29). In comparison, PROs research in oncology has shown defined benefits ranging from improved QoL and reduction in hospitalisations to overall survival (13) and even cost-effectiveness (30).'

Page 20 lines 445-453

'There is a growing body of published, peer-reviewed literature exploring the experiences of both kidney patients and multidisciplinary kidney HCP with ePROs (14, 18, 24, 27, 28, 55). Previous studies have included non-dialysis dependent Chronic Kidney Disease (CKD) and peritoneal dialysis populations (18, 24, 28), evaluating existing ePROs and associated delivery systems. This paper adds to the corpus by using semi-structured interviews to gain rich data to inform and optimise future ePRO implementation in a HD population naïve to ePRO collection and clinicians unused to routine and systematic ePRO use. [

3. Methods

Design

Pragmatism is typically viewed a philosophical approach, not a methodological approach to qualitative work (that I know of). So it makes sense that from a pragmatist orientation, your question guided the design towards a qualitative approach. Did you have a qualitative methodology guiding your study? IE

qualitative description, descriptive exploratory, etc? (Braun & Clarke 2020 seem to locate TA as a method, not a methodology). Please add references.

What is your qualitative research question?

Response: The qualitative research question is: 'What are the experiences and perceptions of patients undergoing HD on the implementation and use of ePROs in routine care and research settings?'

As suggested, within the Methods, the design section of the paper has been redrafted to set out the epistemological and theoretical perspective, as well as the chosen methodology, with references. See tracked changes:

Page 8-9 lines 176-187

'Epistemologically, pragmatism, which is set within a paradigm of enquiry processes and research practicality (34), provides the philosophical framework to answer this question. A core assumption of pragmatism is that research should proceed from a wish to produce actionable knowledge (35); allowing the researcher to select the research design and the methodology deemed most appropriate (36). These foundations led to the decision to use the Consolidated Framework for Interventional Research (37) to offer a theoretical perspective and a Qualitative Descriptive (QD) methodology (38-40). This is particularly useful for healthcare studies focused on discovering the who, what, and where of events or experiences and gaining understanding of inadequately understood phenomenon. Kim et al (40) describe QD as the 'label of choice' when an unambiguous description of a phenomenon is desired or information is sought to develop and refine questionnaires or interventions (41). This choice of methodology led to the utilisation of the following methods:'

Participant selection

Who approached patient participants? (E.G., a researcher, a third party?)

Did home HD and in-centre dialysis patients all come to this one location from which they were invited to participate? In the Results, it reads as if all patients dialyzed at home – so were they all on home HD? And were all HCP working with home dialysis patients? Pretty sure the answer is no give that in Table 1, we read that 15/22 were in-centre, 7/22 HHD – so please just edit paper so this is clear.

Response: The lead researcher first approached patient participants in centre pre-dialysis session and for HHD patients when they attended clinic. The manuscript has been updated to include this detail (**Page 9/line 194-197**):

'Participants were identified and recruited between July 2018 and November 2019 by the lead author. Eligible patients were primarily approached face to face in the dialysis unit; Patients dialysing in-centre were approached pre dialysis session and patients who dialysed at home (HHD) were approached when they attended for clinic.'

Further content has been added to clarify that 15 patients dialysed in-centre and 7 patients dialysed at home (**Page 10, lines 211-214**):

'All participants were being treated via, or working at, a large regional hospital. To accurately reflect the diversity within the dialysis population, patients being treated in-centre (n=15) were recruited from two satellite units as well as 7 patients choosing to dialyse at home.'

In the results, reference is made to HHD patients or patients dialysing in-centre. If there were no divergent opinions then the group is just referred to as 'patients'.

The setting seems to be missing in the methods. Note that the statements in the first paragraph under Results about where participants were treated / worked, recruited etc seems to fit better under Methods than Results.

Response: A 'setting' section has been added to methods and statements referred to above moved to this section. Please see tracked changes **page 10, lines 209-217**:

‘Setting:

Data was collected from 22 patients undergoing haemodialysis and 17 HCPs in the United Kingdom (see tables 1 and 2). All participants were being treated via, or working at, a large regional hospital. To accurately reflect the diversity within the dialysis population, patients being treated in-centre (n=15) were recruited from two satellite units as well as 7 patients choosing to dialyse at home. Non-medical members of the multidisciplinary team were targeted from these two units, one in a city and one serving mainly rural communities. All HCPs currently working in the home setting (n=5) had extensive previous experience of in-centre dialysis delivery.’

Who sent the email of invitation to HCP? (E.G., a manager, clerical staff, a researcher?)

Response: the Lead Researcher sent email of invitation See tracked changes **page 9/line 207-208:** ‘HCPs were recruited from the broader renal team, and initially contacted by email by the lead author.’

Tiny note that HCP is sometimes used, and other times spelled out in full throughout the paper (including in the abstract)

Response: Thank you for flagging this, the manuscript has now been updated accordingly.

Analysis

The use of CFIR is a strength of the study. Does your use of the CFIR assume that the “intervention” being discussed here is about potential use, actual use, or both potential and actual use and implementation of PROs in dialysis care?

Response: For clarification please see **page 11, lines 249-252:**

‘The CFIR is a widely utilised conceptual framework developed to guide systematic assessment of factors that might influence implementation and effectiveness, including assessing potential barriers and facilitators in preparation for implementing an innovation (See Figure 1) (37, 50).’

What does it meant that “participant verification was undertaken on one transcript...” Perhaps clarify and add a reference?

Response: See tracked changes **page 12, line 260:**

‘Participant verification (51), to check that the transcript correctly documented the discussion, was undertaken on one transcript from each group – no discrepant comments were reported.’

Patient Public Involvement – need to put (PPI) behind this to explain acronym

Response: Thank you, the manuscript has been updated as suggested, see **page 12 line 265 ‘Patient and Public Involvement (PPI)’**

4. Results

Might Box 1 be adapted similarly to Box 2 so that both HCP and patient participant exemplar quotes were provided for each key finding and reflect the narrative description in the paper?

Response: Thank you, Box 2 has been adapted as suggested.

Box 2 – “Timing” row. I need help understanding that “horses for courses” means in relation to timing?

Response: Apologies, this is an old English idiom used by a patient – meaning that people are different and will need different approaches. In terms of timing, they were saying some will need a short amount of time to complete, others will need longer. An explanation has been added to box to clarify.

Not sure what “IV” and “I” means before quotes

Response: IV means ‘interviewee’ and I means ‘interviewer’ – now written in full for clarity.

Box 3: A few grammatical mistakes / incomplete sentences; consistent use of quotation marks; use of “PROM” but paper uses “PRO”

Response: Thank you. The manuscript has been corrected. The acronyms PROs or ePROs have been used throughout for consistency.

Finding 3 – it seems contradictive that patients “described a task-oriented style of care...with a focus on practically administering HD...” and that the Home HD team was “responsive and holistic in their approach to care.” Or is it that patient quotes support both ends of this spectrum?

Response: The manuscript has been clarified to demonstrate that 15 participants dialysed in a satellite unit whilst 7 participants undertook home dialysis (in the UK the vast majority of patients dialyse in centre). The approaches to care for these 2 groups was different i.e. task orientated v holistic respectively and was highlighted in the findings as the approach to care affected the way patients perceived the need for change using ePROs.

Finding 4 – a little of this context could be described in Methods/setting that PROs were not used routinely in care, but an annual PREMs were completed (for organizational use?). (This comes back to my earlier point that participants were naïve to PRO use and implementation, and in this study you are then addressing potential future use and implementation of PROs in dialysis care.)

Response: Thank you. This contextual material has been added, see tracked changes **Page 11, lines 231-243:**

‘Since PROs are not routinely collected in the UK, including the hospital trust where this research was undertaken, it was recognised pre-interview information was required to aid the quality of discussion around PROs and future implementation. Therefore participants were provided with a diagram of a core outcome set selected specifically for use in HD trials, to illustrate the position of PROs in outcome measurement (44) and example PROs from a recent systematic review and supported by our local Renal Public and Patient Involvement group: (Kidney Disease Quality of Life SF (KDQOL-SF) (45), Kidney Disease Quality of Life 36 (KDQOL-36) (46) and Integrated Patient Outcome Scale – Renal (IPOS-Renal)) (47). Patients in all UK Renal Units are invited to complete the annual Renal PREM to report their experience of kidney care. This report demonstrates variation in experience across centres and can be used to drive organisational change and improvement (48); some patient participants indicated they had completed the annual PREM.’

5. Discussion

I would suggest that the first statement in the discussion is incorrect. There is a growing body of published, peer-reviewed literature that addresses use of PROs in dialysis care and the experiences of both kidney patients and multidisciplinary kidney HCP. For just a few examples, see PMID: 3519925; 32242823; 26591273; 31632681; 28651528; Smith V, Wise K. Ren Soc Australas J. 2017;13(1):14-21; doi:10.5750/ejpc.v4i2.1125. This literature could then be considered to confirm your findings, or highlight what your study adds that has not previously been articulated.

Response: Thank you, the discussion has been amended accordingly; please see tracked changes **page 20 lines 445-453:**

‘There is a growing body of published, peer-reviewed literature exploring the experiences of both kidney patients and multidisciplinary kidney HCP with ePROs (14, 18, 24, 27, 28, 55). Previous studies have included non-dialysis dependent Chronic Kidney Disease (CKD) and peritoneal dialysis populations (18, 24, 28), evaluating existing ePROs and associated delivery systems. This paper adds to the corpus by using semi-structured interviews to gain rich data to inform and optimise future ePRO implementation in a HD population naïve to ePRO collection and clinicians unused to routine and systematic ePRO use.’

Re. confusion between PROMs and PREMs – please clarify this statement to explain that PREMs are self-reports about individuals' experiences of receiving healthcare -which is distinct from PROs – self-report of individuals' health outcomes relevant to their QoL.

Response: Thank you, this distinction now included in the introduction **page 7 lines 141-145**:
'However, while Patient-Reported Experience Measures (PREMs), which allow patients to self-report their experience of receiving healthcare, have been collected since 2017 (14), PROs are still not routinely and systematically collected to manage individual patient care in the United Kingdom: meaning that many patients and some members of the multidisciplinary team caring for them are inexperienced in PROs and related concepts.'

Additional clarification added in the data collection section **page 11 lines 239-243** for readers unfamiliar with the concepts:

'Patients in all UK Renal Units are invited to complete the annual Renal PREM to report their experience of kidney care. This report demonstrates variation in experience across centres and can be used to drive organisational change and improvement (48); some patient participants indicated they had completed the annual PREM.'

6. Conclusion

This could be strengthened to provide the reader with your "final take-home" message from your study.

Response: The conclusion has been rewritten to address this, see tracked changes **page 24, lines 538-550**:

'To conclude the SARs-COV-19 pandemic has caused an irreversible shift in healthcare delivery, with increased use of digital communication and assessment. While the nephrology community, both patients and HCPs, are largely supportive of the concept of ePROs, there remain caveats to their routine and systematic use. Stakeholders need to be convinced by empirical evidence, considering the best available measures and methodological considerations. By exploring the experiences, views and perceptions of major stakeholders, this study identified key factors at organisational and individual levels which would assist effective implementation of ePROs. Further research will then be required to evaluate any subsequent ePROs interventions to empirically demonstrate impact and benefit of their use to the dialysis community.'

Supplementary files

Table 1. # invited / # participating = 22/41 – but I'm wondering if these numbers are reversed and it should be 41 were invited and 22 participated? Same comment for Table 2 re healthcare professionals

Response: Thank you for flagging. You are correct – the manuscript has been updated accordingly.

Tables 1 & 2 – do the black "x's" mean that this was coded in NVIVO? (whereas grey "x's" denote first coding in NVIVO)

Response: You are correct – please see the key on original document (original documents were in colour and grey x was red).

Response to Reviewer 2:

Many thanks for taking the time to review this paper.

This qualitative paper focuses on the use of PROMs during management of patients with kidney disease on HD. The topic is relevant, and is timely considering covid and the need for alternative approaches to patient management.

The use of a pragmatic approach is appropriate, and there is considerable discussion of reflexivity, which is often lacking in qualitative papers such as this.

My only comment would be that the interview questions are very closed and, in some cases, a little leading (e.g. should a 'good' clinician already be able to elicit those outcomes that are important to a patient?), which is very similar to the quote listed in Box 2 under Form of Feedback. Was the question added to the topic guide after this interview? Or is this quote in direct response to the question?

I appreciate that the questions need to be focused, but I think it would be important to acknowledge the task-focused nature of the questions in the discussion, and to make clear the context of this and any other quotes that were in response to direct questions such as this.

Response: The topic guide did develop iteratively after piloting and use. The guide was used as a prompt and questions were not read verbatim or necessarily in the order they appeared in the topic guide, to maintain the semi-structured nature of the interview.

VERSION 2 – REVIEW

REVIEWER	Andrea Gibbons University of Winchester Department of Psychology
REVIEW RETURNED	02-Jul-2021

GENERAL COMMENTS	The paper is much improved and is suitable for publication.
---